# Differentiation of *Penicillium roqueforti* from Closely Related Species Contaminating Cheeses and Dairy Environment

**Miloslava Kavková** [1,2,*], **Jaromír Cihlář** [1,2], **Vladimír Dráb** [1] and **Ladislav Bár** [1,2]

1   Depatment of Cheese Technologies, Dairy Research Institute, Ltd., Ke Dvoru 12a,
    160 00 Praha, Czech Republic; j.cihlar@vum-tabor.cz (J.C.); v.drab@vum-tabor.cz (V.D.);
    l.bar@vum-tabor.cz (L.B.)
2   Culture Collection of Dairy Microorganisms, Milcom, Ltd., Ke Dvoru 12a, 160 00 Praha, Czech Republic
*   Correspondence: m.kavkova@vum-tabor.cz; Tel.: +420-23-679-012

**Abstract:** Currently, *Penicillium roqueforti* and the closely related *P. carneum* and *P. paneum* are identified based on their macromorphology, micromorphology, and molecular properties, the determination of which involves time-consuming procedures. Culture collections focused on dairy isolates of *P. roqueforti* require quick and efficient tools for routine applications to identify the (a) taxonomy affiliation and (b) morphological properties of strains that influence the sensory properties of blue-veined cheeses. Here, we assessed the morphological variability of *P. roqueforti*, *P. carneum*, *P. paneum,* and *P. crustosum* on artificial, Edam-like, and Roquefort-like media. Molecular tools were used to test *P. roqueforti* strains and clones effectively. A novel primer, PrsF, was tested for specificity within strains and isolates of *P. roqueforti* compared to *P. carneum*, *P. paneum*, and *P. crustosum*. The results reveal that PrsF was specific to the *P. roqueforti* samples and did not amplify the other tested *Penicillium* species. Identification based simultaneously on the specificity of the PrsF primer pair and cultivation of *P. roqueforti* strains on Roquefort-like medium represents an effective method for expanding the collections and practical use of *P. roqueforti* in the dairy industry.

**Keywords:** contaminants; *Penicillium roqueforti*; phenotypic variability; specific marker





## 1. Introduction

The filamentous fungus *Penicillium roqueforti* Thom is employed in the cheese manufacturing industry as a secondary starter culture to produce blue-veined cheeses.

Various *P. roqueforti* strains are deposited worldwide in national collections of microorganisms, collections belonging to the food industry, and small private collections. These strains typically originate from organic substrates, such as compost, silage, and moulded food, and spaces traditionally related to the production of blue-veined cheese, e.g., cellars, lofts, and other areas in dairies. The origin of *P. roqueforti* strains plays a significant role in their technological, biochemical, and morphological properties [1]. *P. roqueforti*, *P. carneum*, and *P. paneum* belong to the *P. roqueforti* group [2,3]. *P. carneum* and *P. paneum* can often cause undesired contaminations in culture collections and be responsible for the mistaken identification of isolates during routine analyses. The other two members of the *P. roqueforti* group, *P. psychrosexualis* [4] and *P. mediterraneum* [5], have never been isolated from the dairy environment and cheeses. Nowadays, the cheese industry operates under the HACCP (Hazard Analysis Critical Control Points) system, which prevents and controls cheese production against food-borne diseases, contaminants, and their vectors. Despite this fact, cheese manufacture is not a sterile process, and contamination with fungal spores is possible in many ways.

The routine identification of *P. roqueforti* strains is based on the macromorphology of colonies on media, such as Creatin Sucrose Agar (CREA), Malt Extract Agar (MEA), Czapek Yeast Agar (CYA), and Yeast Extract Sucrose Agar (YES), and the temperature of fungal cultures. This characterisation is only partially comparable with a description of

typological CBS strains documented by [6,7], owing to the morphological variability of strains belonging to the *P. roqueforti* group. The classification of the same strains based on their microscopical characteristics requires expert technical skills to distinguish micromorphological structures using light and electron microscopy [6]. Various strains of *P. roqueforti* present macromorphological and micromorphological variabilities in vitro. The intraspecific variability of *P. roqueforti*, based on macromorphological features and confirmed using molecular methods, was highlighted in recent studies [1,8].

The specific physiological properties of *P. roqueforti*, such as colour and smell, as well as lipolytic and proteolytic activities, are necessary for cheese production. These properties can only be evaluated using specific diffusion tests on media and cheese slurry with corresponding supplements [9,10].

In general, the molecular techniques routinely used to identify fungal strains are effective and accurate. The Internal Transcribed Spacer region (ITS) is the most common molecular marker for fungi at the genus and species levels [11,12]. However, the DNA sequencing of fungal strains based only on the ITS region has been reported to show inaccuracy in identification at the species level, particularly for closely related species; a combination of ITS with secondary identification markers, such as β-tubulin (*BenA*), calmodulin (*CaM*), elongation factor (*TEF-1α*), and RNA polymerase II second largest subunit (RPB2) gene, is required to identify *Penicillium* spp. accurately at the species level [13–15].

Supplementary genomic methods such as RAPD analysis or microsatellite markers are effective for phylogenetic studies [2,3]. However, this method requires experimental skills to obtain adequate reproducibility associated with their implementation, precluding their routine application in culture collections.

Although the studies mentioned above have identified strains in the *P. roqueforti* group using macromorphology, micromorphology, and molecular tools, there remains a need for simple and easy-to-perform techniques in routine identification processes. Accordingly, this study aimed to explore the intraspecific variability of *P. roqueforti* strains on artificial media (MEA, CYA, YES, CREA) and two cheese-based media against three contaminants, namely *P. carneum*, *P. paneum*, and *P. crustosum*. The variability detected within the *P. roqueforti* strains prompted the development of an optimised genetic marker specific to *P. roqueforti* species. This marker was successfully amplified by PCR and tailored to exclude or confirm the presence of DNA from the best known closely related contaminating species such as *P. carneum* and *P. paneum*, as well as other fungal contaminants belonging to the *Penicillium* genus, in the tested samples.

## 2. Materials and Methods

### 2.1. Strain Cultivation and Morphological Features

Fungal strains originated from the culture collections CCDM and CCDBC (Milcom Ltd., Tábor, Czech Republic), as well as CCF (Charles University, Prague, Czech Republic) (Table 1). The macromorphology of strains was evaluated using a three-point position on agar plates containing CYA, YES, CREA, MEA, and on cheese-based media with Edam and Roquefort, as described in Table 2. The colour, texture, margin, and diameter of fungal colonies were determined after 7 days of culture at 25 °C. The Munsell Soil Colour Charts were used to assess the colours of the obverse and reverse parts of each selected colony. Colony diameter was determined after 7 days of culture at 25 °C using a central-point position, with each colony measured twice. For each strain and clone used in this study, 20 isolates were measured and evaluated for variability using the ANOVA statistical software (version 12.1.) (StatSoft Europe, Hamburg, Germany). The microscopical examination was performed under a light microscope (Olympus BX43, Olympus Corporation, Tokyo, Japan) at magnifications of 400×, 600×, and 1000×. Conidia, phialides, stipes, conidiophores, rami, and metulae were stained with Lactophenol blue solution (Sigma-Aldrich, Darmstadt, Germany) and subsequently measured. On an average, we

obtained 50 measurements of each assessed structure for each strain and clone using the (QuickPhotoCamera 3.2, PROMICRA, Prague, Czech Republic).

**Table 1.** Strains of *Penicillium* spp. used in this study (access date 1 June 2021).

| Strains | Origin | Accession No. |
|---|---|---|
| CCDM 296 Penicillium roqueforti | Culture Collection of Dairy Microorganisms, Milcom Ltd., Czech Republic | MW600465 |
| CCDM 294 +clones a, b, c, d Penicillium roqueforti | Culture Collection of Dairy Microorganisms, Milcom Ltd., Czech Republic | MW600466, MW600467, MW600468, MW600469, MW600470 |
| CCF 3837 Penicillium carneum | Culture Collection of Fungi, Dpt. Botany, Fac. Biology, Charles University, Prague, Czech Republic | MW600471 |
| CCF 3839 Penicillium paneum | Culture Collection of Fungi, Dpt. Botany, Fac. Biology, Charles University, Prague, Czech Republic | MW600472 |
| CCDBC 304 Penicillium crustosum | Culture Collection of Dairy and Bakery Contaminants, Milcom Ltd., Czech Republic | MW600473 |
| CCDBC 331 Penicillium solitum | Culture Collection of Dairy and Bakery Contaminants, Milcom Ltd., Czech Republic | MW600474 |
| CCDBC 311 Penicillium chrysogenum | Culture Collection of Dairy and Bakery Contaminants, Milcom Ltd., Czech Republic | MW600475 |
| CCDBC 303 Penicillium rubens | Culture Collection of Dairy and Bakery Contaminants, Milcom Ltd., Czech Republic | MW600476 |
| CCDBC 312 Penicillium glabrum | Culture Collection of Dairy and Bakery Contaminants, Milcom Ltd., Czech Republic | MW600477 |
| CCDBC 307 Penicillium expansum | Culture Collection of Dairy and Bakery Contaminants, Milcom Ltd., Czech Republic | MW600478 |

**Table 2.** Cultivation media used for the macromorphological description of *Penicillium* sp.

| Medium Acronym | Medium Description | Origin | Cultivation | References |
|---|---|---|---|---|
| MEA | Malt Extract Agar | MERCK KGaA, Germany | 25 °C | |
| CREA | Creatine Sucrose Agar | MILCOM a.s.; Czech Republic | 25 °C | Samson et al., 2010 [7] |
| YES | Yeast Extract Sucrose Agar | MILCOM a.s.; Czech Republic | 25 °C | |
| CYA | Czapek Yeast extract Agar | MILCOM a.s.; Czech Republic | 30 °C | |
| | **Cheese-like media** | | | |
| Edam-like | A total of 278 g grated Eidam cheese after salting, 630 g Cream 32% fat, 91.3 g Reconstituted skim milk, 10 g NaCl, 10 g Trisodium citrate dihydrate, 10 g Tripotassium citrate monohydrate, 12 g agar, 250 mL UHT milk 3.5% fat, homogenization at 85 °C for 10 min in Thermomix (Vorwerk CS, Prague, Czech Republic) | | 25 °C | Modified Hansen and Nielsen, 1996 [16] |
| Roquefort-like | A total of 200 g grated Roquefort cheese after salting, 30 g NaCl, 10 g agar, tap water to 1000 mL, homogenization by Ultra-Turrax T25 at 9500 min$^{-1}$ (Janke-Kunkel GmbH, Staufen, Germany), sterilization 102 °C for 20 min | | 25 °C | |

## 2.2. P. roqueforti-Specific Primer Design

The specific primer was designed based on multiple sequence alignment of the ITS region with members of *Penicillia* from different sections of the genus [15,17]. The ITS region was chosen deliberately because, unlike the other routinely used barcodes, its sequences are homologous in different *P. roqueforti* strains. ITS sequences (Table 3) were downloaded from the NCBI database and ISHAM barcoding database websites (http://www.ncbi.nlm.nih.gov/ (accessed on 7 October 2021); http://its.mycologylab.org/ (accessed on 7 October 2021)).

**Table 3.** Strains of *Penicillium* spp. used to design the specific primer.

| Strains | Section | Accession No. |
|---------|---------|---------------|
| *Penicillium roqueforti* | *Roquefortorum* | EU427296 [1], MITS3851 [2], MITS3852 [2], MITS3853 [2] |
| *Penicillium carneum* | *Roquefortorum* | HQ442338 [1] |
| *Penicillium paneum* | *Roquefortorum* | HQ442346 [1] |
| *Penicillium psychrosexualis* | *Roquefortorum* | HQ442345 [1] |
| *Penicillium mediterraneum* | *Roquefortorum* | NR172393 [1] |
| *Penicillium crustosum* | *Fasciculata* | AF033472 [1] |
| *Penicillium solitum* | *Fasciculata* | AY373932 [1] |
| *Penicillium chrysogenum* | *Chrysogena* | AF033465 [1] |
| *Penicillium rubens* | *Chrysogena* | JX997057 [1] |
| *Penicillium glabrum* | *Aspergiloides* | GU981567 [1] |
| *Penicillium expansum* | *Penicillium* | AY373912 [1] |
| Source database | [1] NCBI http://www.ncbi.nlm.nih.gov/ (accessed on 7 October 2021) [2] ISHAM http://its.mycologylab.org/ (accessed on 7 October 2021) | |

All sequences were aligned using the MUSCLE algorithm [18], and the generated data were visually inspected for variable sites within the homologous region (Figure 1). The most variable region in *P. roqueforti* with a significantly different sequence, compared to other *Penicillia*, in the alignment was selected as a template for the ITS-Pr primer specific to *P. roqueforti*. Using the OligoAnalyzer Tool (https://www.idtdna.com/ (accessed on 7 October 2021)), the preferred oligonucleotide was next analysed for potential hairpin loops as well as self- and hetero-dimers, with ITS4 as the intended reverse primer.

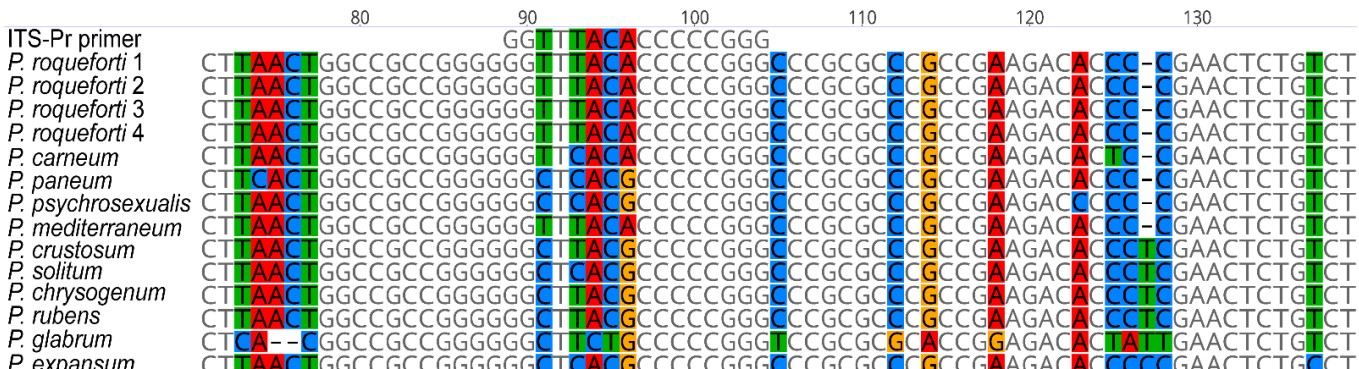

**Figure 1.** Alignment data generated when designing the primers specific to *P. roqueforti* DNA. Sequences of the ITS regions of *P. roqueforti*, *P. carneum*, *P. paneum*, *P. psychrosexualis*, and *P. mediterraneum* (section *Roquefortorum*); *P. crustosum*, *P. solitum* (section *Fasciculata*); *P. chrysogenum*, *P. rubens* (section *Chrysogena*); *P. glabrum* (section *Aspergiloides*); and *P. expansum* (section *Penicillium*) were aligned using the MUSCLE algorithm to produce homologous regions. The variable site that showed a different nucleotide sequence of *P. roqueforti* to the other *Penicillia* was selected to design the primer sequence.

2.2.1. DNA Extraction from Mycelial Samples

Total DNA was obtained from four phenotypically different strains of *P. roqueforti* (CCDM296 and CCDM294), four distinct clones derived from the original CCDM294 culture (294-a, 294-b, 294-c, and 294-d), *P. carneum* (CCF3837), *P. paneum* (CCF3839), and *P. crustosum* (CCDBC304). These strains and clones were deposited in CCDM, CCDBC, and CCF collections. DNA was extracted from mycelial samples via a rapid method using NaOH. Using a sterile loop, a portion of the mycelium was removed and placed into prepared microtubes containing a small amount of glass beads (G1145, Sigma-Aldrich, Darmstadt, Germany); 20 μL of 0.5 M of NaOH was then added to each vial, and the latter was transferred to a Mo Bio vortex adapter. After securing the adapter to the vortex, the samples were homogenised in microtubes at the maximum power for 5 min, followed by centrifugation at 13,000× *g* for 2 min. Supernatants were collected in clean microtubes and diluted 10-fold with 10 mM Tris-Cl (pH 8.5).

DNA was quantified using Qubit® Fluorometer 3.0 and the dsDNA HS assay kit (both from Thermo Fisher Scientific, Eugene, OR, USA). Total DNA extracts were stored at −20 °C and further diluted, as required, for subsequent PCR amplification.

2.2.2. ITS and β-Tubulin Sequencing

The ITS region of nuclear ribosomal DNA (ITS1, 5.8S rDNA, and ITS2) of all the studied samples was amplified using the ITS1f (5′-CTT GGT CAT TTA GAG GAA GTA A-3′) and ITS4 (5′-TCC TCC GCT TAT TGA TAT GC-3′) primers [19]. Beta-tubulin was amplified as a secondary barcode using the BT2a (5′-GGT AAC CAA ATC GGT GCT GCT TTC-3′) and BT2b (5′-ACC CTC AGT GTA GTG ACC CTT GGC-3′) primers [20]. PCR was performed using in a 25 μL reaction volume [0.5 μL of each primer (200 nM), 1 μL of DNA template (100 ng), 12.5 μL of 2× PPP Taq MasterMix (TopBio, Prague, Czech Republic), and 10.5 μL of ddH$_2$O)]. The amplification conditions were as follows: pre-heating at 95 °C for 2 min, 35 cycles each of denaturation at 95 °C for 30 s, annealing at 55 °C (ITS primers) and 58 °C (tubulin primers), respectively, for 30 s, and extension at 72 °C for 60 s, followed by a final extension at 72 °C for 8 min.

PCR products were separated on a 1% agarose gel (SeaKem® LE; Lonza, Rockland, ME, USA) stained with GelRed® (Biotium, Fremont, QC, Canada) at 70 V for 60 min, and visualised using the GENE GENIUS Bio Imaging System (Syngene, Frederick, MD, USA). All PCR products were treated with 2 μL of ExoSAP-IT[TM] according to the manufacturer's reference manual (ThermoFisher Scientific, Baltic UAB, Vilnius, Lithuania), and sequenced by Eurofins Genomics Germany GmbH (Ebersberg, Germany) using a suitable forward primer.

*2.3. ITS-Pr Primer Testing*

Specificity of the ITS-Pr primer was tested via PCR using ITS4 as the reverse primer. PCR was carried out using 1 ng of DNA from each *P. roqueforti* strain or clone (CCDM 294, 294-a, 294-b, 294-c, 294-d, and CCDM 296), *P. carneum* (CCF 3837), *P. paneum* (CCDBC 3839), and *P. crustosum* (CCDBC 304). *P. carneum* and *P. paneum* were deliberately selected as the closest species related sequentially and morphologically to *P. roqueforti*. *P. crustosum* was chosen as the common contaminant. PCR was performed in a 25 μL reaction volume (0.5 μL of each primer (200 nM), 1 μL of DNA template (100 ng), 12.5 μL of 2× PPP Taq MasterMix (TopBio, Czech Republic), and 10.5 μL of ddH$_2$O). The amplification conditions were as follows: pre-heating at 95 °C for 2 min, 35 cycles of denaturation at 95 °C for 30 s, annealing at 52 °C for 30 s, and extension at 72 °C for 60 s, followed by a final extension at 72 °C for 8 min. PCR products were separated and visualised according to the method described in Section 2.2.2.

## 3. Results

### 3.1. Morphological Features

3.1.1. Macromorphology

After 7 days of cultivation at 25 °C, the studied strains displayed macromorphological diversity on agar-based and cheese-based media. Variabilities in the diameters (mm) of colonies and the macromorphological features of the tested strains are described in detail in Tables A1–A6 and Figure 2.

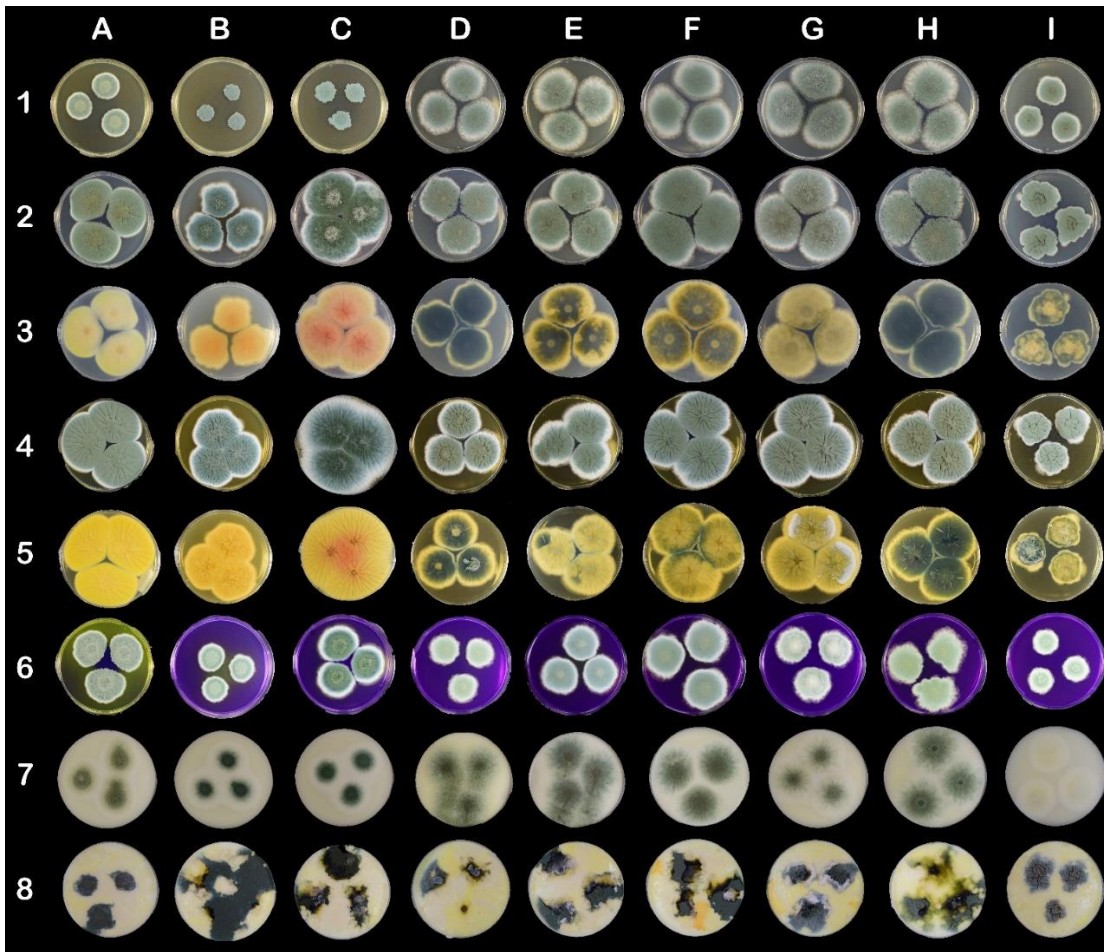

**Figure 2.** Morphological variability on mycological and cheese-like media on Petri dishes (90 mm). The columns represent strains: (A) CCDBC 304 (*P. crustosum*), (B) CCF 3839 (*P. carneum*), (C) CCF 3837 (*P. paneum*), (D) CCDM 296 (*P. roqueforti*), (E) CCDM 294 and clones (F) 294a, (G) 294b, (H) 294c, and (I) 294d (*P. roqueforti*). The rows represent mycological and cheese-like media: (1) MEA, (2) CYA top side, (3) CYA bottom side, (4) YES top side, (5) YES bottom side, (6) CREA, (7) Roquefort-like medium, (8) Edam-like medium.

All the *P. roqueforti* strains, except clone 294d, showed intensive growth on MEA medium. Clone 294d produced velvety umbonate irregular colonies, whereas the remaining strains and clones produced flat velutinous colonies (Table A1; Figure 2 (1D–1I)). The diameters of *P. crustosum*, *P. paneum*, and *P. carneum* colonies were significantly smaller than those of *P. roqueforti* colonies (excluding clone 264d). Additionally, clone CCDM 294d was also significantly smaller in diameter on CYA medium (<30 mm) (Table A2) than other strains belonging to *P. roqueforti*. The diameters of *P. crustosum* (CCDBC 304) and *P. paneum* (CCF 3837) colonies exceeded 36 mm. The morphological variability within the strains and clones of *P. roqueforti* was apparent, particularly on the reverse side of cultures (Table A2, Figure 2 (3D–3I)). Although *P. roqueforti* colonies were pale blue-green on the obverse side, the variability in colour and texture on the reverse side was evident. Clone 294d produced velvety umbonate irregular small colonies, whereas the remaining *P. roqueforti* strains and

clones produced colonies that were flat, filamentous, and partially fasciculate on CYA (Table A2; Figure 2 (2D–2I).

Extensive growth and sporulation of all the strains were observed on YES medium (Table A3). Diameters (mm) of the largest colonies were determined for *P. paneum* and *P. crustosum*. Colonies of *P. roqueforti* CCDM 296 and clone CCDM 294d were significantly small in diameter (mm). All colonies were considerably wrinkled on the obverse and reverse sides.

On CREA medium, clone CCDM 194d produced small velutinous colonies with irregular wrinkles and fasciculate in the centre (Table A4; Figure 2 (6I)). The remaining strains and clones of *P. roqueforti* generated velutinous colonies with radially aligned wrinkles. A different colony pattern was also noted on the CREA medium. Furthermore, discolouration of the CREA medium inoculated with *P. crustosum* and *P. carneum* was noted.

The growth, colonisation pattern, and characteristics of colonies produced on Edam-like media (Figure 2, line 8) were different to those growing on Roquefort-like media (Figure 2, line 7). Edam-like media generated irregularly shaped colonies. The colonies of *P. roqueforti* strains and clones varied in colour, form, and elevation. They were rugose or flat and formed by dense mycelia. Clone 294d produced rugose colonies with the smallest diameter (Table A5). *P. crustosum*, *P. paneum*, and *P. carneum* grew extensively on Edam-like media. *P. roqueforti* colonies growing on Roquefort-like medium differed from those growing on Edam-like medium, as they produced sparse, powdery mycelia. Margins of the colonies were creamy white and comprised 1/2–1/3 of the colony diameter (Table A6). Clone 294d produced yellowish-white colonies on Roquefort-like medium, whereas other clones generated dark yellowish-green colonies.

### 3.1.2. Micromorphology

The sizes of micromorphological structures within *P. roqueforti* strains and clones did not show any significant variability. The textures of these structures were also similar and consistent with those reported in the literature. The only variability was noted in the colour of conidia, particularly in clone 264d. The micromorphological descriptions of *P. paneum*, *P. carneum*, and *P. crustosum* corresponded with data presented in the literature.

### 3.2. Verification of the Taxonomic Affiliation of Target Species and Strains via PCR

The taxonomic affiliation of the tested species and strains was assessed by identifying sequences of the generated ITS region using the BLAST alignment tool. To further verify the taxonomic classification of the studied samples, β-tubulin was successfully sequenced as a secondary barcode. The sequences were uploaded to the NCBI database. The accession numbers are listed in Tables 1 and A7. The success rate of PCR amplification and the sequencing results further confirm that the quality and purity of the extracted total DNA were sufficient for all subsequent applications.

### 3.3. Application of ITS-Pr Primer

We used the whole DNA sequence amplified by ITS primers to identify the only suitable annealing site for ITS-Pr, which is variable for most species in the alignment, but particularly important for *P. roqueforti* when compared to other *Penicillia*. The ITS-Pr annealing site is situated in the internal transcribed spacer 1 region (ITS1) 89 bp from the start of the NRRL:749 type strain ITS sequence (EU427296) and spans the next 16-bp 5′-GGT TTA CAC CCC CGG G-3′ sequence (Figure 1). The ITS-Pr primer is limited to 16 bp because of the high GC content of the surrounding DNA sequence. Despite its length, the ITS-Pr annealing site contains up to five variable positions in the aligned sequences.

The ITS-Pr primer was tested for its ability to anneal and amplify DNA of *P. roqueforti* strains (CCDM 296, CCDM 294, and 294a–d). The closely related *P. carneum* (CCF 3837), *P. paneum* (CCF 3839), and *P. crustosum* (CCDBC 304), which represent common contaminating species, were included in the analysis as the control. The primer tests revealed that the primer pair ITS-Pr–ITS4 was specific enough to amplify and detect *P. roqueforti* samples

exclusively and excluded the remaining *Penicillia* from amplification (Figure 3). The ITS region was also amplified under the same conditions, using suitable primers and appropriate annealing temperatures. The resulting products served as positive controls for all tested *Penicillia*, confirming the presence of particular genomic DNA in all ITS-Pr–ITS4 PCR reactions (Figure 3).

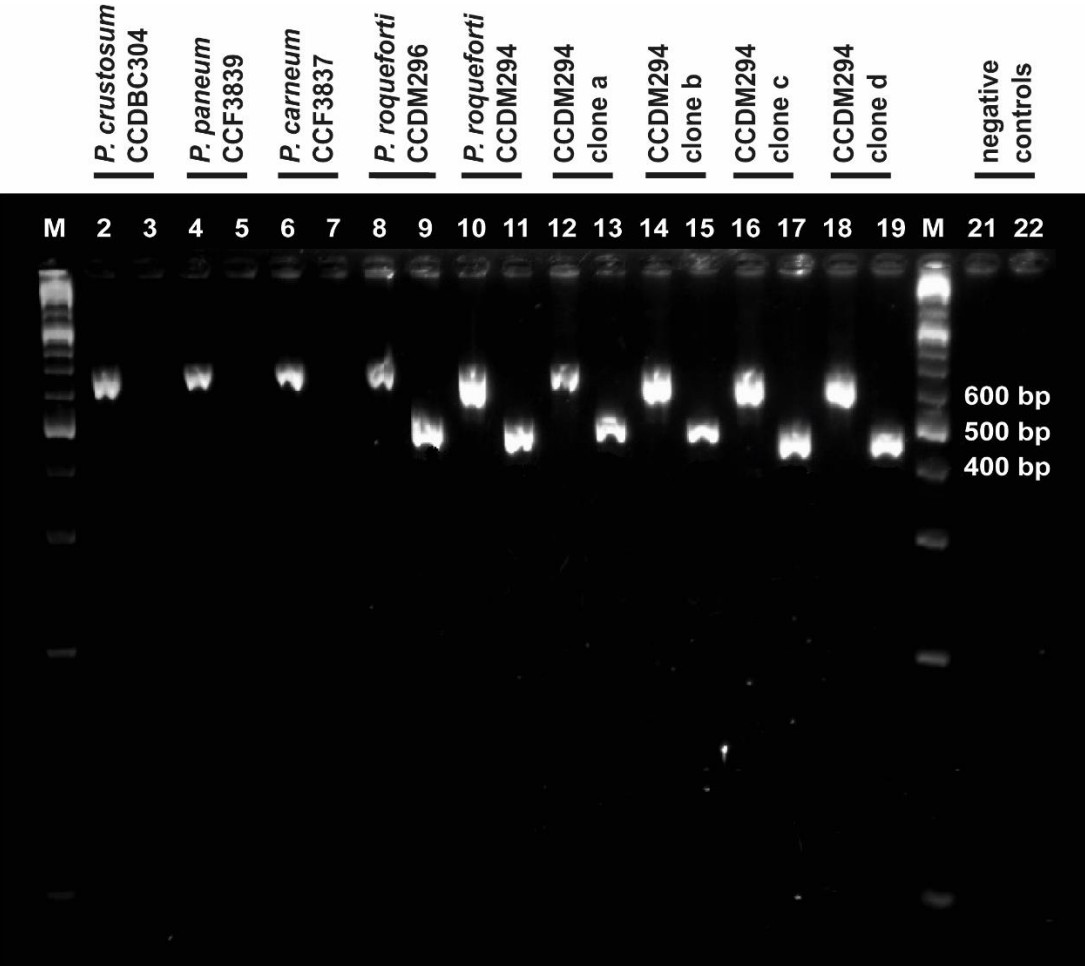

**Figure 3.** Determination of ITS-Pr specificity. Bands in lanes with even numbers represent PCR products of the ITS1f-4 reaction and confirm each gDNA sample (suitably purified and diluted). Bands in lanes with odd numbers represent PCR products of the ITS-Pr–ITS4 reaction; these were present in *P. roqueforti* samples, but absent in *P. carneum* (lane 7), *P. paneum* (lane 6), and *P. crustosum* (lane 5). Lanes 1 and 20 are labelled as M and represent the GeneRuler™ molecular marker (Fermentas). The negative controls for ITS1f–ITS4 and ITS-Pr–ITS4 were loaded in lanes 21 and 22, respectively. The positions of DNA bands correspond with their expected sizes.

## 4. Discussion

The experimental manipulation of *P. roqueforti* strains in collections includes the periodic revitalisation of strains, control of viability and germinability, as well as the acquisition of any new isolates. The testing of additional properties related to practical applications in the dairy industry requires specific media [16,21]. In the present study, media were designed to simulate the cheese slurry of Edam- and Roquefort-type cheeses.

Although *P. paneum*, *P. carneum*, and *P. crustosum* displayed different patterns on mycological media, it was virtually impossible to distinguish among the strains of *P. roqueforti* on an Edam-like medium. The verification of strains on the Roquefort-like medium was evident only when compared with macromorphology on mycological media, micromorphology, and genetic analysis. Clone 294d produced a different pattern of colonisation on

all tested mycological media. The 294d colonies on Edam-like media were similar to those of other *P. roqueforti* strain and clones. The difference was visible only when compared with other *P. roqueforti* strains on Roquefort-like medium. The significant discolouration of the colonies on Roquefort-like media suggested that this clone was not suitable for manufacturing blue-veined cheese. Moreover, the different pattern produced on mycological media might result in misidentification [3,8].

Species that are closely related to *P. roqueforti* and some contaminants belonging to *Penicillium* spp. grow in a pattern similar to that of *P. roqueforti*, particularly on cheese-based media. The macromorphological variability of *P. roqueforti* strains and clones reflect the unpredictable intraspecific variability of *P. roqueforti* isolates [1,8,22]. This occurrence can lead to confusion and the contamination of collections during routine procedures, if no other analyses are performed. The macromorphological description of *P. roqueforti* on mycological media [7], combined with microscopical and molecular analyses, is effective only for the characterisation of new acquisitional isolates.

Species-specific primers are commonly used to routinely identify certain fungal species [23]. Such primers are generally designed to target variable regions of conserved genes—either protein-coding genes [24,25] or ribosomal RNA genes [26,27]. The ITS region of rRNA, in particular, was used to design ITS-Pr. Our findings revealed that up to five variable positions within the annealing site of ITS-Pr were sufficient to differentiate between *P. roqueforti* strains and other contaminating *Penicillia*, including the closely related species belonging to the *Roqueforti* group [15,17]. Particularly, we were able to discriminate between *P. roqueforti* and *P. carneum*, the sequences of which differ only by one substitution of T for C (Figure 1) within the annealing site. Owing to this substitution, the ITS-Pr–ITS4 primer pair did not recognise, nor allowed for amplification of the *P. carneum* sequence, despite the relatively low annealing temperature used (Figure 3). The specificity of the ITS-Pr primer could theoretically be improved by introducing a mismatch at the 3′ end [28–30], i.e., moving the variable region of the primer to the 3′ end. However, during the experimental testing, the specificity of the designed primer proved to be sufficient and therefore no further modifications were needed. Our study does not include two species from the *Roqueforti* group, namely *P. mediterraneum* [5], and *P. psychrosexualis* [4], because none of them have yet been isolated from cheeses and the dairy environment. Therefore, we could not test the compatibility of the ITS-Pr primer with their DNA experimentally. The two species were isolated in very different environments from those in dairy spaces and dairy products. The first mentioned was isolated from dung in Spain and was described as a new species only recently [5]. The second was obtained from a wooden crate in a cold store of apples in the Netherlands, and its growth is restricted to temperatures lower than 20°C [4]. However, as deduced from the alignment (Figure 1), the ITS sequences corresponding to the annealing site of ITS-Pr primer are identical in *P. psychrosexualis* and *P. paneum*. Additionally, as the ITS-Pr primer did not recognise the *P. paneum* sequence, we can assume that *P. psychrosexualis* would also not be amplified. In the case of *P. mediterraneum*, the result of the sequence comparison is completely different. The new *Penicillium* species has been described mostly based on sequence differences in *BenA*, *CaM*, and RPB2 barcodes. ITS sequences of *P. roqueforti* strains and *P. mediterraneum* are identical, making both species phylogenetically the closest [5]. Currently, the PCR using the ITS-Pr primer is a very potent tool to rapidly differentiate *P. roqueforti* from common contaminating *Penicillium* species, including the closely related ones, and allows for routine identification of *P. roqueforti* strains in dairy and dairy culture collections. In the case that *P. mediterraneum* is proved as associated with dairy products, the additional barcodes will need to be sequenced. With respect to the origin of *P. mediterraneum*, the HACCP system in dairies provides sufficient control.

## 5. Conclusions

The cultivation of *P. roqueforti* strains on media simulating certain types of cheese (Roquefort-like medium) media showed the technological suitability of the strain for

cheese manufacture [16]. The cultivation of dairy contaminants from the *P. roqueforti* group on cheese-like media showed that these three contaminants are hardly distinguishable from *P. roqueforti* in routine practice. The implementation of the PCR method based on species-specific primers is a suitable method for routinely identifying a selected species against others. Such primers are generally designed based on the variable regions of the conserved genes among several inter-genus species [25,26,31]. Our primer was designed to target the ITS region, which is the most common marker in fungal phylogeny. Although ITS sequences are not always sufficient to accurately identify isolates at the species level [12,13,21] we have successfully identified a region of DNA in the ITS barcode that is unique to *P. roqueforti* strains compared to *P. carneum*, *P. paneum*, and *P. crustosum*, representing dairy contaminants. This region facilitated the designing of a primer that showed exclusive annealing to *P. roqueforti* DNA. Our results reveal that the ITS-Pr primer is accurate and specific enough to identify *P. roqueforti*, and could be used as a rapid tool to detect contamination in dairy and cheese production.

**Author Contributions:** Conceptualization, M.K., J.C. and V.D.; methodology, M.K., J.C. and V.D.; software, J.C. and L.B.; validation, J.C., M.K. and V.D.; formal analysis, M.K. and J.C.; investigation, V.D.; resources, L.B.; data curation, J.C. and M.K.; writing—original draft preparation, M.K. and J.C.; writing, M.K. and J.C.; visualization, L.B. and J.C.; supervision, M.K. and V.D.; project administration, M.K. and V.D.; funding acquisition, M.K. and V.D. All authors have read and agreed to the published version of the manuscript.

**Funding:** This research was funded by THE MINISTRY OF AGRICULTURE OF THE CZECH REPUBLIC, MZE-RO1418, THE NATIONAL AGENCY FOR AGRICULTURAL RESEARCH OF CZECH REPUBLIC, grant numbers QK1910036 and QK1910024, and The National Programme on Conservation and Utilization of Plant, Animal and Microbial Genetic Resources Important for Food and Agriculture (NPGR) belonging to the Ministry of Agriculture of the Czech Republic.

**Institutional Review Board Statement:** Not applicable.

**Informed Consent Statement:** Not applicable.

**Data Availability Statement:** The data that support the findings of this study are present in the Appendix A.

**Acknowledgments:** We would like to acknowledge the CCF collection, Charles University, Prague for providing cultures of *P. paneum* and *P. carneum*.

**Conflicts of Interest:** All the authors declare that they have no relevant conflict of interest. The funders had no role in the design of the study; in the collection, analyses, or interpretation of data; in the writing of the manuscript, or in the decision to publish the results.

## Appendix A

**Table A1.** Colony morphology and diameter (mm) of *Penicillium* sp. strains on MEA at 25 °C (7 days). * The data represent the average diameter of fungal colony ± s.e. ANOVA, $F_{(8,171)}$ = 213.8; $p \leq 0.05$ ($n$ = 20). Index letters assign significant differences. CCDBC 304 (*P. crustosum*), CCF3839 (*P. carneum*), CCF 3837 (*P. paneum*), CCDM 296 (*P. roqueforti*), CCDM 294, and clones 294 a, b, c, d (*P. roqueforti*).

| Strain/Clone | Diameter (mm) * | Figure Position | Colony Texture | Margin | Colour | |
|---|---|---|---|---|---|---|
| | | | Line 1 | | | Munsell |
| **CCDBC 304** | 26.5 [d] ± 1.22 | A | Velutinous, irregular, flat | Arachnoid, creamy white, regular | Pale green, yellow green in the centre | 5G 7/2 |
| **CCF 3839** | 25.6 [d] ± 0.88 | B | Velutinous colony, rugose, partially wrinkled (2/3), irregular | Arachnoid, undulate | Pale blue | 5B 6/2 |
| **CCF 3837** | 25.8 [d] ± 1.15 | C | Velutinous colony, irregular, partially fasciculate in centre | Arachnoid, undulate | Pale blue | 5B 6/2 |

**Table A1.** *Cont.*

| Strain/Clone | Diameter (mm) * | Figure Position | Colony Texture | Margin | Colour | |
|---|---|---|---|---|---|---|
| | | Line 1 | | | | Munsell |
| **CCDM 296** | 36.4 [ab] ± 0.46 | D | Velutinous colony | Arachnoid, creamy white, regular | Pale blue green | 5BG 7/2 |
| **CCDM 294** | 38.3 [a] ± 0.72 | E | Velutinous partially fasciculate in centre, flat, | Arachnoid, creamy white, irregular | Pale blue green | 5BG 7/2 |
| **Clone 294a** | 42.1 [a] ± 1.15 | F | Velutinous flat | Arachnoid, creamy white, regular | Pale blue green | 5BG 7/2 |
| **Clone 294b** | 41.6 [a] ± 1.54 | G | Velutinous partially fasciculate, flat, | Arachnoid, creamy white, regular | Pale blue green | 5BG 7/2 |
| **Clone 294c** | 38.41 [a]± 0.81 | H | Velutinous partially fasciculate, flat, | Arachnoid, creamy white, regular | Pale blue green | 5BG 7/2 |
| **Clone 294d** | 27.8 [d] ± 0.32 | I | Velutinous, umbonate, irregular | Arachnoid, creamy white, irregular | Pale green | 10G 6/2 |

**Table A2.** Colony morphology and diameter (mm) of *Penicillium* sp. strains on CYA medium at 25 °C (5 days). * The data represent the average diameter of fungal colony ± s.e. ANOVA, post hoc LSD test, $F_{(8,171)}$ = 116.4; $p \leq 0.05$ ($n$ = 20). Index letters assign significant differences. CCDBC 304 (*P. crustosum*), CCF 3839 (*P. carneum*), CCF 3837 (*P. paneum*), CCDM 296 (*P. roqueforti*), CCDM 294, and clones 294 a, b, c, d (*P. roqueforti*).

| Strain/Clone | Diameter * (mm) | Figure Position | Colony Texture | Margin | Colour | | | |
|---|---|---|---|---|---|---|---|---|
| | | Line 2, 3 | | | Upper Side | Munsell | Reverse | Munsell |
| **CCDBC 304** | 36.98 [e] ± 0.38 | A | Velutinous, rugose, wrinkles from centre to edges, regular | Creamy white, regular | Greyish blue green | 5BC 5/2 | Yellow, slight wrinkles | 5Y 9/10 |
| **CCF 3839** | 25.25 [d] ± 0.32 | B | Velutinous, fasciculate in the centre, exudate droplets, irregular | Irregular, white | Moderate blue | 5B 5/6 | Yellow, slight wrinkles | 5Y 9/14 |
| **CCF 3837** | 36.70 [e] ± 0.34 | C | Velutinous, flat, fasciculate in the centre, wrinkled, exudates, irregular | Creamy white, regular | Pale blue green | 5BG 7/2 | Yellow with pale orange in centre, wrinkles, cracks | 7 YR 7/8 |
| **CCDM 296** | 32.95 [c] ± 0.38 | D | Velutinous, partially filamentous, flat, irregular | White, irregular, arachnoid | Pale blue green | 5BG 7/2 | Dusky yellowish greenYellow margins | 10GY 3/2 |
| **CCDM 294** | 31.72 [c] ± 0.32 | E | Velutinous, partially filamentous fasciculate in the centre | White, irregular, arachnoid | Pale blue green | 5BG 7/2 | Dusky yellowish greenYellow intercalary | 10GY 3/27.5Y 7/164 |
| **Clone 294a** | 34.76 [ce] ± 0.23 | F | Velvety, fasciculate in the centre | Creamy white, regular | Pale blue green | 5BG 7/2 | Dusky yellowish greenYellow intercalary | 10GY 3/27.5Y 7/164 |
| **Clone 294b** | 30.6 [b] ± 0.32 | G | Velutinous, partially filamentous, fasciculate, flat | White regular thin | Pale blue green | 5BG 7/2 | Moderate yellow | 5Y 7/6 |
| **Clone 294c** | 29.78 [b] ± 0.28 | H | Velutinous, partially filamentous, sparse, flat | White irregular thin | Pale blue green | 5BG 7/2 | Dusky yellowish greenYellow— only margins | 10GY 3/27.5Y, 7/164 |

**Table A3.** Colony morphology and diameter (mm) of *Penicillium* sp. strains on YES medium at 25 °C (5 days). * The data represent the average diameter of fungal colony $\pm$ s.e. ANOVA, post hoc LSD test, $F_{(8,171)} = 369.3$; $p \leq 0.05$, ($n = 20$). Index letters assign significant differences. CCDBC 304 (*P. crustosum*), CCF 3839 (*P. carneum*), CCF 3837 (*P. paneum*), CCDM 296 (*P. roqueforti*), CCDM 294, and clones 294 a, b, c, d (*P. roqueforti*).

| Strain/Clone | Diameter (mm) * | Figure Position | Colony Texture | Margin | Colour | | | |
|---|---|---|---|---|---|---|---|---|
| | | Line 4, 5 | | | Upper Side | Munsell | Reverse | Munsell |
| **CCDBC 304** | 37.2 [d] ± 0.8 | A | Velutinous, rugose, regular radial wrinkles | White, regular | Greyish green | 10GY 5/2 | Yellow, radial wrinkles | 7.5Y 9/10 |
| **CCF 3839** | 29.7 [e] ± 0.8 | B | Velutinous, rugose, irregular wrinkles and cracks | White, regular | Pale blue | 5B 6/2 | Yellow, radial wrinkles, cracks | 5Y 8/16 |
| **CCF 3837** | 37.8 [d] ± 0.8 | C | Velutinous, rugose, weaved wrinkles radial, dark exudates in the centre | White, regular | Pale blue green | 5BG 7/2 | Yellow, radial rippled wrinkles, orange centre | 5Y 8/16 7.5R 8/10 |
| **CCDM 296** | 20.4 [a] ± 0.8 | D | Velutinous, rugose, weaved wrinkles radial and concentric | White, irregular, arachnoid | Pale blue green | 5BG 7/2 | Dusky yellowish green Yellow margin | 10GY 3/2 7.5Y 7/164 |
| **CCDM 294** | 25.4 [b] ± 0.8 | E | Velutinous, rugose, weaved wrinkles radial and concentric | White, irregular, arachnoid | Pale blue green | 5BG 7/2 | Yellow, radial wrinkles, cracks | 7.5Y 9/6 |
| **Clone 294a** | 32.3 ± 0.8 | F | Velutinous, rugose, weaved wrinkles radial | White, irregular, arachnoid | Pale blue green | 5BG 7/2 | Dusky yellowish green Yellow, radial wrinkles Mycelia intercalary | 7.5Y 9/8 |
| **Clone 294b** | 32.1 [c] ± 0.8 | G | Velutinous, rugose, weaved wrinkles radial | White, irregular, arachnoid | Pale blue green | 5BG 7/2 | Yellow, radial wrinkles, cracks, intergrow white mycelia | 7.5Y 9/6 |
| **Clone 294c** | 26.7 [b] ± 0.8 | H | Velutinous, rugose, weaved wrinkles radial, irregular, short | White, irregular, arachnoid | Pale blue green | 5BG 7/2 | Dusky yellowish green, yellow margin cracks radially | 10GY 3/2 7.5Y 7/164 |
| **Clone 294d** | 19.4 [a] ± 0.8 | I | Velutinous, rugose, weaved wrinkles, short cranks, concentric, irregular, short | White, irregular, arachnoid | Pale blue | 5B 6/2 | Dusky yellowish green margin, yellow weaved cracks irregular | 10GY 3/2 7.5Y 7/164 |

**Table A4.** Colony morphology and diameter (mm) of *Penicillium* sp. strains on CREA. * The data represent the average diameter of fungal colony $\pm$ s.e. ($n = 20$), ANOVA, post hoc LSD test, $F_{(8,171)} = 302.9$; $p \leq 0.05$. CCDBC 304 (*P. crustosum*), CCF 3839 (*P. carneum*), CCF 3837 (*P. paneum*), CCDM 296 (*P. roqueforti*), CCDM 294, and clones 294 a, b, c, d (*P. roqueforti*). Index letters represent significant differences.

| Strain/Clone | Diameter (mm) * | Figure Position | Colony Characteristics | Margin | Colour | |
|---|---|---|---|---|---|---|
| | | Line 6 | | | | Munsell |
| **CCDBC 304** | 22.4 [b] ± 0.6 | A | Velvety, circular form, small droplets, flat intensive discoloration of CREA | Irregular arachnoid white | Very pale green | 10G 5/2 |

**Table A4.** *Cont.*

| Strain/Clone | Diameter (mm) * | Figure Position | Colony Characteristics | Margin | Colour | |
|---|---|---|---|---|---|---|
| | | Line 6 | | | | Munsell |
| **CCF 3839** | 13.4 [a] ± 0.8 | B | Velvety, circular form, fasciculate in the centre, discoloration in the centre only | Irregular arachnoid white | Light blue green | 5BG 5/2 |
| **CCF 3837** | 23.1 [b] ± 0.3 | C | Velvety, circular form, droplets in the centre, flat | Irregular arachnoid white | Light green to light blue green | 5G 7/4 5BG 5/2 |
| **CCDM 296** | 17.4 [a] ± 1.2 | D | Velvety, circular form, fasciculate in the centre, flat | Irregular arachnoid white | Very pale green | 10G 5/2 |
| **CCDM 294** | 21.6 [b] ± 0.5 | E | Velvety, circular form, fasciculate in the centre, flat | Irregular arachnoid white | Pale blue | 5B 6/2 |
| **Clone 294a** | 23.8 [b] ± 0.4 | F | Velvety, circular form, fasciculate in the centre, flat | Irregular arachnoid white | Light blue green | 5BG 6/6 |
| **Clone 294b** | 20.2 [ab] ± 0.6 | G | Velvety, circular form, fasciculate in the centre, umbonate | Irregular arachnoid white | Pale green | 10G 6/2 |
| **Clone 294c** | 22.4 [b] ± 1.2 | H | Velvety, irregular form, fasciculate in the centre, flat | Irregular arachnoid white | Moderate yellowish green | 10GY 6/4 |
| **Clone 294d** | 17.5 [a] ± 0.2 | I | Velvety, circular form, fasciculate in the centre, flat | Irregular arachnoid white | Very pale green | 10G 5/2 |

**Table A5.** Colony morphology and diameter (mm) of *Penicillium* sp. strains on Edam-like medium. * The data represent the average diameter of fungal colony ± s.e. ($n$ = 20), ANOVA, post hoc LSD test, $F_{(8,171)}$ = 228.3; $p \leq 0.05$. Index letters represent significant differences. CCDBC 304 (*P. crustosum*), CCF 3839 (*P. carneum*), CCF 3837 (*P. paneum*), CCDM 296 (*P. roqueforti*), CCDM 294, and clones 294 a, b, c, d (*P. roqueforti*).

| Strain/Clone | Diameter (mm) * | Figure Position | Colony Characteristics | Margin | Colour | |
|---|---|---|---|---|---|---|
| | | Line 8 | | | | Munsell |
| **CCDBC 304** | 21.8 [b] ± 0.9 | A | Velutinous, regular form, raised | Irregular, blue green | Moderate blue green | 5BG 4/6 |
| **CCF 3839** | 24.5 [c] ± 0.3 | B | Velutinous, irregular form, wavy pattern | Irregular, blue green | Dusky blue green | 5BG 3/2 |
| **CCF 3837** | 24.4 [c] ± 0.5 | C | Velutinous, irregular form, rugose, flat | Irregular, green, indistinct | Dusky yellowish green | 10GY 3/2 |
| **CCDM 296** | 22.5 [bc] ± 0.8 | D | Velutinous, irregular form, rugose, aerial mycelia | Irregular, blue-green, indistinct | Pale blue green | 5BG 7/2 |
| **CCDM 294** | 20.8 [b] ± 0.8 | E | Velutinous, irregular form, rugose, secondary white mycelia | Irregular, greyish green, indistinct | Greyish green | 10G 4/2 |
| **Clone 294a** | 21.6 [b] ± 1.4 | F | Velutinous, irregular form, flat, exudate droplets | Irregular, green | Greyish green | 10G 4/2 |
| **Clone 294b** | 24.7 [c] ± 1.2 | G | Velutinous, irregular form, rugose, secondary aerial mycelia | Irregular, white-blue shaggy | Dusty blue green | 5 BG 3/2 |
| **Clone 294c** | 23.6 [c] ± 0.5 | H | Velutinous, irregular form, umbonate | Irregular, arachnoid | Greyish green | 10G 4/2 |
| **Clone 294d** | 14.3 [a] ± 1.2 | I | Velutinous, irregular form, rugose | Irregular, white thin | Greyish blue | 5 PB 5/2' |

**Table A6.** Colony morphology and diameter (mm) of *Penicillium* sp. strains on Roquefort-like medium. * The data represent the average diameter of fungal colony ± s.e. (*n* = 20), ANOVA, post hoc LSD test, $F_{(8,171)}$ = 15.15; $p \leq 0.05$. Index letters assign significant differences. CCDBC 304 (*P. crustosum*), CCF 3839 (*P. carneum*), CCF 3837 (*P. paneum*), CCDM 296 (*P. roqueforti*), CCDM 294, and clones 294 a, b, c, d (*P. roqueforti*).

| Strain/Clone | Diameter (mm) * | Figure Position | Colony Characteristics | Margin | Colour | |
|---|---|---|---|---|---|---|
| | | Line 7 | | | | Munsell |
| **CCDBC 304** | 26.49 [c] ± 0.21 | A | Fasciculate, regular flat form | White regular | Moderate yellowish green | 10GY6/4 |
| **CCF 3839** | 24.51 [a] ± 0.34 | B | Velutinous, regular flat form | White regular | Moderate blue green | 5BG 4/6 |
| **CCF 3837** | 24.24 [d] ± 0.52 | C | Velutinous, powdery, regular flat form | White regular | Pale blue green | 5BG 7/2 |
| **CCDM 296** | 25.21 [a] ± 0.21 | D | Powdery, irregular flat form | Yellow, arachnoid, irregular | Dark yellowish green | 10 GY 4/4 |
| **CCDM 294** | 26.03 [a] ± 0.32 | E | Powdery, regular flat form | White irregular, 1/3 of whole colony diameter | Greyish green | 10G 5/2 |
| **Clone 294a** | 28.5 [c] ± 0.32 | F | Powdery, regular flat form | Creamy, arachnoid, irregular | Greyish blue green | 10G 5/2 |
| **Clone 294b** | 24.09 [b] ± 0.32 | G | Powdery, regular flat form | Regular, white, 1/2 of whole colony diameter | Dark yellowish green | 10GY 4/4 |
| **Clone 294c** | 24.67 [a] ± 0.32 | H | Powdery, irregular flat form | White irregular, 1/3 of whole colony diameter | Dark yellowish green | 10GY 4/4 |
| **Clone 294d** | 25.45 [a] ± 0.32 | I | Powdery, regular flat form, fasciculate in the centre | White regular | Yellowish gray | 5Y 8/1 |

**Table A7.** Accession numbers of β-tubulin (*BenA*) sequences from *Penicillium* spp. used in this study.

| Strains | Accession No. |
|---|---|
| CCDM 296 *Penicillium roqueforti* | MZ520338 |
| CCDM 294 +clones a, b, c, d *Penicillium roqueforti* | MZ520339, MZ520340, MZ520341, MZ520342, MZ520343 |
| CCF 3837 *Penicillium carneum* | MZ520344 |
| CCF 3839 *Penicillium paneum* | MZ520345 |
| CCDBC 304 *Penicillium crustosum* | MZ520346 |
| CCDBC 331 *Penicillium solitum* | MZ520347 |
| CCDBC 311 *Penicillium chrysogenum* | MZ520348 |
| CCDBC 303 *Penicillium rubens* | MZ520349 |
| CCDBC 312 *Penicillium glabrum* | MZ520350 |
| CCDBC 307 *Penicillium expansum* | MZ520351 |

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
