# Peer review of "Differentiation of Penicillium roqueforti from Closely Related Species Contaminating Cheeses and Dairy Environment"

_fermentation, doi:10.3390/fermentation7040222_

Round 1

Reviewer 1 Report

First of all, I would like comment that I found very interesting the approach of the article.

I am trying to make some suggestions and qualify the text.

First, you make reference in the manuscript to Table 2 and Table 3 (lines 88 and 106, respectively). Those tables are nowhere to be found.
In this regard, I would like a brief description of how the cheese-based media has been prepared (it could be in table 2, but it does not appear).
In addition, in the appendix are Table 2A and Table 4A which are not discussed in the text.
Lines 167 to 170 the text has a different size.
Line 205 use a homogeneous nomenclature with the rest of the text: change (Figure 2 (I6) to (Figure 2 (6I), first the number and then the letter.
Finally, comment that, in my view, I think that using a little more bibliography would enrich both the discussion and the description of the results obtained.
With these small changes I hope the final manuscript will be improved and will help its publication.

Author Response

Dear Sir,

Thank you for your time spent reviewing our manuscript. Your thoroughgoing control reveals some mistakes and the absence of informative facts.  We improved the manuscript according to your recommendations.  

First of all, we supplemented tables 2 and 3. Table 2 includes the detailed description of the preparation both of cheese media. We added references to tables A2 and A4 into the section Results. The size of the text (Lines 167-170) was unified, and the figure description was improved. We enriched the discussion about adequate comments and bibliography to increase the clarity of the manuscript. Additionally, we unified the titles of tables in the appendix - the strains acronyms. hope that our corrections are everything you expected and that we did away with all the deficiencies. 

Sincerely,

Dr. Miloslava Kavková

Dairy Research Institute Ltd.

Ke Dvoru 12a

Prague 6, 160 00

Czech Republic

Reviewer 2 Report

Kavková et al. propose an elegant PCR-based method for the identification of Penicillium roqueforti from other closely related species which are difficult to distinguish by physiological test and morphological observations, hence the article has the potential to be of great interest for the dairy industry scientists. The article is concisely and clearly written in an academic grade English, however, I have some remarks I will list below. The introduction describes the interest in the topic with a brief enumeration of the currently used methods for P. roqueforti identification. The materials and methods section describes the methods used for the study in a reproducible manner. The results section is clearly written and visualized by figures with acceptable graphic quality. The discussion section is profound and adequate to the obtained results.

Even tough my impression of the article is positive and I will recommend it for publication within the MDPI Fermentation, I have several remarks to be addressed by the authors:

Line 17 – “verify” should be replaced by “test”;

Lines 62-63 – The sentence should be split in two after references [13-15] and before “Combination of ITS…”

Lines 67-69 – I cannot agree RAPD analysis is time-consuming and costly, however it requires experimental skills to obtain reproducibility

Line 78 – “during PCR” should be replaced by “by PCR”

Lines 144–147 and lines 164-166 – it is appropriate to describe the PCR mixes by giving the reaction volume and the final concentrations of the components and not by microliters

Lines 235-240 – it will be preferable to discuss a little bit further the motifs for the positioning of the ITS-Pr primer, especially why not using a mismatching for the other species 3’-base which will greatly improve the specificity (at least in theory)

Lines 278-284 – The paragraph is a little bit redundant repeating parts of the introduction section

Lines 295-301 – I believe the focus of this section should be on the fact that P. mediterraneum and P. psychrosexualis were never isolated from a dairy product and not on the fact they were unavailable which sounds not very serious…

Lane 307 – “And as” should be replaced by “As”

Lane 312 – The primer could not be a tool, the PCR reaction with the primers could be a tool.

Author Response

Dear Sir,

Thank you for your time spent reviewing our manuscript. We appreciated your remarks point to crucial parts of the manuscript. 

According to your comments, we managed the following changes:

  • Line 17 we replaced the “verify” by “test”
  • Line 62-63 - we split the sentences and references
  • Lines 67-69 - the sentence was changed
  • Line 78 - “during PCR” was changed “to PCR”  
  • Lines 144-147 and 164 -166 - the final concentration of the components were added for PCR reaction
  • Lines 235-240 - the positioning of the ITS-Pr primer is discussed
  • Lines 295-301 - the sentence were modified 
  • Line 307 - “and as” were replaced by As
  • Line 312 - the sentence was changed in the appropriate version 

Other minor revisions were done. We supplemented tables 2 and 3.  Table 2 includes the detailed description of the preparation both of cheese media. We added references to tables A2 and A4 into the section Results.

We corrected the form of acronyms in tables A1-A6 in the appendix. Four new references and citations were filled into Discussion”, “Conclusion”, and “References”. 

I hope that our corrections are everything you expected and that we did away with all the deficiencies. 

Sincerely,

Dr. Miloslava Kavková

Dairy Research Institute Ltd.

Ke Dvoru 12a

Prague 6, 160 00

Czech Republic